# White Spot, a Novel Endoscopic Finding, May Be Associated with Acid-Suppressing Agents and Hypergastrinemia

**DOI:** 10.3390/jcm10122625

**Published:** 2021-06-15

**Authors:** Noriko Nishiyama, Hideki Kobara, Maki Ayaki, Shintaro Fujihara, Kaho Nakatani, Naoya Tada, Kazuhiro Koduka, Takanori Matsui, Tadayuki Takata, Taiga Chiyo, Nobuya Kobayashi, Tingting Shi, Koji Fujita, Joji Tani, Tatsuo Yachida, Tsutomu Masaki, Ken Haruma

**Affiliations:** 1Department of Gastroenterology, Nishiyama Neurological Hospital, Sakaide City 762-0023, Kagawa, Japan; 2Department of Gastroenterology and Neurology, Faculty of Medicine, Kagawa University, MiKi City 761-0793, Kagawa, Japan; kobara@med.kagawa-u.ac.jp (H.K.); joshin@med.kagawa-u.ac.jp (S.F.); nakatanikaho@gmail.com (K.N.); n-tada@med.kagawa-u.ac.jp (N.T.); koduka2525@gmail.com (K.K.); tk-matsui@med.kagawa-u.ac.jp (T.M.); t-takata@med.kagawa-u.ac.jp (T.T.); t_chiyo@med.kagawa-u.ac.jp (T.C.); nobuyak@med.kagawa-u.ac.jp (N.K.); shitingtingc@med.kagawa-u.ac.jp (T.S.); 92m7v9@med.kagawa-u.ac.jp (K.F.); georget@med.kagawa-u.ac.jp (J.T.); tyachida@med.kagawa-u.ac.jp (T.Y.); tmasaki@med.kagawa-u.ac.jp (T.M.); 3General Medical Center, Department of General Internal Medicine 2, Kawasaki Hospital, Kawasaki Medical School, Okayama City 700-8505, Okayama, Japan; kmnb1979@yahoo.co.jp (M.A.); kharuma@med.kawasaki-m.ac.jp (K.H.)

**Keywords:** endoscopy, gastric mucosa, white spot, antacids, gastrinemia, white globe appearance, acid-suppressing agent, potassium-competitive acid blocker, proton-pump inhibitor

## Abstract

White globe appearance (WGA) is defined as a microendoscopic white lesion with a globular shape underlying the gastric epithelium and is considered a marker of gastric cancer. We recently reported that endoscopically visualized white spot (WS) corresponding to WGA appeared on the nonatrophic mucosa of patients with acid-suppressing agents (A-SA) use. We evaluated patients undergoing routine esophagogastroduodenoscopy and divided the patients into an A-SA group (*n* = 112) and a control group (*n* = 158). We compared the presence of WS in both groups. We also compared WS-positive- (*n* = 31) and -negative (*n* = 43) groups within the A-SA group regarding these patients’ backgrounds and serum gastrin concentrations. Comparing the A-SA group with controls, the prevalence of WS was significantly higher (31/112 vs. 2/158; *p* < 0.001). The number of patients with high serum gastrin concentrations was significantly higher in the WS-positive group (18/31) vs. the WS-negative group (5/43) (*p* < 0.001). Within the A-SA group, the prevalence of WS was also significantly higher in patients taking potassium-competitive acid blockers vs. proton-pump inhibitors (21/31 vs. 10/31, *p* < 0.001). The WS-positive group had a significantly greater percentage of patients, with a high serum gastrin level (*p* < 0.001). WS may be associated with hypergastrinemia and potassium-competitive acid blockers.

## 1. Introduction

White globe appearance (WGA) is characterized as a microendoscopic white lesion with a globular shape (<1 mm) underlying the gastric epithelium, with the presence of overlying microvessels. WGA is microscopically detected by magnifying endoscopy and is reported to be an endoscopic marker that is highly specific in differentiating early gastric cancer [1,2]. WGA is also found in autoimmune atrophic gastritis [3,4]. We recently found a novel macroendoscopic finding called white spot (WS), which corresponds to microendoscopic WGA in noncancerous and autoimmune gastritis. We also reported that the presence of WS may be related to potassium-competitive acid blocker (P-CAB) uptake and hypergastrinemia [5] in the gastric mucosa without cancerous lesion and autoimmune gastritis.

Acid-suppressing agents (A-SAs) include proton-pump inhibitors (PPIs) and P-CAB, which is a more powerful antacid. These A-SA are commonly available for the treatment of functional gastrointestinal disorders and acid-related diseases, such as peptic ulcers and reflux esophagitis. However, long-term use of PPIs causes a number of adverse effects, including bone fracture [6], *Clostridium difficile* infection [7], pneumonia [8], and atrophic gastritis [9], and has the potential to cause gastric cancer [10] or gastric carcinoid [11] due to hypergastrinemia. Furthermore, PPI use causes various endoscopic gastric mucosal changes, such as fundic gland polyps (FGP) [12], white flat elevated mucosa [13], black spot [14], gastric cobblestone-like mucosa (GCSM), and gastric cracked mucosa (GCM) [15] because of parietal cell degeneration. Therefore, we hypothesized that WS might be one of these antacid-related findings. In the present study, we aimed to investigate the clinicopathological features of WS and its relationship with A-SA usage.

## 2. Materials and Methods

### 2.1. Study Design, Setting, and Participants

This single-institution observational study was conducted at the Nishiyama Neurological Hospital (Sakaide, Kagawa, Japan). All consecutive patients who underwent routine esophagogastroduodenoscopy (EGD) between March 2020 and August 2020 were included.

The patients were divided into two groups: those using an A-SA (A-SA group) with PPI or P-CAB uptake and those not using an A-SA (control group). Patients using other antacids (including H2-blockers) were included in the control group because these agents are not strong A-SAs and our previous experience has shown that no WS appeared in patients taking these antacids. The prevalence of WS was compared between the two groups. Furthermore, for patients within the A-SA group with available blood testing data, the background characteristics, serum gastrin level, and pathological findings were compared between the WS-positive and WS-negative groups. Blood testing was performed at the patients’ discretion. Patients in the A-SA group who refused to undergo blood testing were excluded from this analysis. The study was approved by the Clinical Ethics Committee of our hospital. All patients provided written informed consent to undergo the procedures and participate in the study.

### 2.2. Evaluation of Clinical Findings

The following clinical characteristics were recorded: sex, age, oral antacid usage, serum anti-*Helicobacter pylori* antibody level, renal function, serum gastrin level, and type of A-SA. Patients who had taken an oral antacid for more than 1 month were defined as antacid users. The anti-*H. pylori* antibody level was measured with the LZ test (Eiken Latex kit; Eiken Chemical Industries, Tokyo, Japan). In accordance with the manufacturer’s instructions, the antibody titer was defined as positive for *H. pylori* at a cutoff value of ≥10 U/mL. Renal function was evaluated using the estimated glomerular filtration rate (eGFR) as normal to moderate (eGFR ≥ 45 mL/min/1.73 m^2^; stage G1–G3a) or moderate to severe (eGFR < 45 mL/min/1.73 m^2^; stage G3b–G5). The serum gastrin level was measured with a Gastrin-RIA kit II (FUJIREBIO Corp, Tokyo, Japan) in the fasting period before the endoscopic examination.

### 2.3. Definition of White Spot

The endoscopic diagnosis of WS was made in accordance with the following procedures. First, when a white substance appeared in white light imaging (Figure 1a), we judged whether the substance was composed of deposits by performing water washing. Second, we evaluated the presence of microvascularity on the white substance, using magnifying narrow-band imaging (Figure 1b). Finally, when the white substance exhibited the characteristic findings of WGA, it was confirmed as WS.

### 2.4. Evaluation of Endoscopic Findings

EGD was performed by a single expert endoscopist (N.N.) who had performed endoscopic examinations in more than 10,000 patients. All endoscopic images obtained from each patient were reviewed by two expert endoscopists (N.N. and M.A.).

EGD was performed with the patient awake or under sedation with midazolam. EGD was carried out using a single-channel video endoscope (GIF-H260Z or GIF-H290Z; Olympus, Tokyo, Japan). Carbon dioxide insufflation was used during the EGD. PPI-related gastric lesions were also recorded.

### 2.5. Histopathological Investigation

Biopsy specimens obtained from gastric WS areas were stained with hematoxylin and eosin. The presence or absence of parietal cell protrusion (PCP), parietal cell vacuolation (PCV), and a dilated gland that contained eosinophilic material with necrotic epithelial fragments were determined. The histological diagnosis was performed by experienced pathologists who were experts in the gastrointestinal field.

### 2.6. Outcome Measures

The primary outcome measure was the comparison of the prevalence of WS between the A-SA and control groups. The secondary measures were the comparisons of the background characteristics and serum gastrin levels between the WS-positive and WS-negative groups composed of patients within the A-SA group for whom blood testing data were available.

### 2.7. Statistical Analysis

Continuous variables were presented as the mean (standard deviation). Differences in categorical variables between two groups were examined by Fisher’s exact test when required. Continuous variables were compared using Welch’s t-test. *p* < 0.05 was considered statistically significant. A multivariate analysis was performed using multiple logistic regression to detect a causative factor of WS prevalence. All statistical analyses were conducted using JMP 15.1.0 (SAS Institute Inc., Cary, NC, USA).

## 3. Results

### 3.1. Study Flowchart

A total of 271 patients underwent EGD during the study period. Of these 271 patients, one patient was diagnosed with autoimmune gastritis and was excluded from the present study. Consequently, a total of 270 patients were assessed. The A-SA group comprised 112 patients, while the control group comprised 158 patients. The study flowchart is shown in Figure 2.

### 3.2. Characteristics of the 270 Patients in the Control and A-SA Groups

The clinical features and endoscopic findings of the patients are shown in Table 1. The male/female ratio did not significantly differ between the control group (57/101) and the A-SA group (49/63; *p* = 0.209). However, the mean age was significantly younger in the control group (61.8 years old) than in the A-SA group (78.7 years old; *p* < 0.001).

The prevalence of WS was 27.6% (31/112) in the A-SA group and 1.3% (2/158) in the control group.

### 3.3. Prevalence of WS in Each PPI Group versus the P-CAB Group within the A-SA Group

Within the A-SA group, the prevalence of WS was significantly lower in those who used a PPI (10/81, 12%), compared with those who used P-CAB (21/31, 68%; *p* < 0.001) (Table 2).

### 3.4. Patient Characteristics in the A-SA Group

Of the 112 patients in the A-SA group, 38 were excluded because they did not provide informed consent for blood testing. Accordingly, the 74 patients included in the A-SA group comprised 31 in the WS-positive group and 43 in the WS-negative group. There were no significant differences between the WS-positive and WS-negative groups in sex, age, detection of *H. pylori*, and renal function. However, the fasting serum gastrin level was significantly higher in the WS-positive group (mean 624 ng/L, range 95–10,082 ng/L) than the WS-negative group (mean 261 ng/L, range 73–954 ng/L; *p* < 0.001). When a serum gastrin level of ≥500 ng/L was defined as high, the percentage of patients with a high serum gastrin level was significantly greater in the WS-positive group (18/31, 58%) than the WS-negative group (6/43, 14%; *p* < 0.001). The percentage of P-CAB users was significantly higher in the WS-positive group (21/31, 68%) than the WS-negative group (5/43, 12%; *p* < 0.001). These results are summarized in Table 3.

Within the group of 26 patients who used P-CAB, the median serum gastrin level was compared between WS-positive patients (*n* = 21) and WS-negative patients (*n* = 5). The serum gastrin level was significantly higher in the WS-positive patients (mean 624 ng/L, range 289–3150 ng/L) than the WS-negative patients (mean 374 ng/L, range 121–954 ng/L; *p* = 0.032) (Figure 3). Furthermore, the serum gastrin level was significantly higher in the WS-positive group than in the groups with other PPI-related gastric lesions (Figure 4).

Multiple logistic regression analysis of WS prevalence was shown in Table 4. The multivariate analysis revealed that high gastrin and P-CAB user were independent causative factors of WS prevalence with an odds ratio of 6.099 (*p* = 0.0224) and 25.73 (*p* = 0.0004), respectively.

### 3.5. Characteristics of Endoscopic Findings

The endoscopic features of WS in 33 patients (31 in the A-SA group, and two in the control group) are summarized in Table 5. In all patients, WS appeared only on the fundic gland areas. The prevalence of WS was highest in the upper gastric body (79%), followed by the middle (48%) and lower (24%) bodies of the stomach. Multiple WSs (two or more) were present in 88% of patients (29/33) (Figure 5a). Nonatrophic mucosa surrounded by WS was observed in 93% of patients (31/33). Nonatrophic mucosa with a round pit surrounded by a white spot was confirmed using magnifying narrow-band imaging (Figure 5b). In the WS-positive group, the observed PPI-related gastric lesions were black spot in 70% (23/33), FGP in 15% (5/33), white flat elevated mucosa in 46% (15/33), GCSM in 27% (9/33), and hyperplastic polyps in 2% (2/33). An example of GCSM accompanied by WS is shown in Figure 5c.

### 3.6. Histopathological Findings

Gastric WS was biopsied in 14 patients. A dilated gland that contained eosinophilic material with necrotic epithelial fragments was found in 6 of 14 patients (43%) (Figure 6a); all dilated glands were crypt epitheliums. PCP and PCV (Figure 6b) appeared in 71% (10/14) and 86% (12/14) of patients, respectively (Table 6).

## 4. Discussion

This observational study revealed three important findings, namely, (1) the prevalence of WS was higher in patients who used an A-SA, (2) the WS-positive group had a significantly greater prevalence of a high serum gastrin level and a higher percentage of P-CAB users than the WS-negative group, and (3) histopathological examination confirmed a high prevalence of PCP and PCV in biopsy samples of WS.

WGA was first described as a supportive marker for the accurate diagnosis of early gastric cancer [1,2]. WGA appears on the demarcation line around cancer and is considered a pooled necrotic substance. A previous study showed that the prevalence of WGA is significantly greater in patients with gastric cancer (15/70, 21.4%) than in patients with noncancerous lesions (3/118, 2.5%; *p* < 0.001); the three WGA-positive noncancerous lesions were benign open ulcer, gastritis, and a low-grade adenoma with an ulcer scar [2]. Although two-thirds of these cases had accompanying ulceration, the cause of WGA remains unknown. Moreover, this previous study did not report antacid use and serum gastrin level. We recently found a novel macroendoscopic finding, termed WS, which is concordant with microendoscopic WGA in noncancerous and nonautoimmune stomachs [5]. To the best of our knowledge, the present study is the first to investigate the relationship between WS and A-SA usage.

In the present study, the PPI-related gastric lesions in WS-positive patients comprised black spot in 70%, FGP in 15%, white flat elevated mucosa in 46%, and GCSM in 27%. A previous study reported that the prevalences of GCM and black spots are 10% (3.7% in the non-PPI group, and 24.4% in the PPI group) and 6.2%, respectively [13,15]. Another study reported that an *H. pylori* post-eradication status is a significant risk factor for the appearance of black spots [16]. Furthermore, the reported prevalence of white flat elevated mucosa is 4.9% (3.3% in the non-PPI group, and 14.3% in the PPI group) [13]. The prevalences of these PPI-related gastric lesions are similar to the prevalence of WS (12%) shown in the present study. Furthermore, in the present study, the prevalence of WS was significantly higher in the A-SA group (27.6%) than in the control group (1.3%). Thus, WS may be a novel endoscopic finding of A-SA-related gastric lesions.

As shown in Table 2, the prevalence of WS was significantly lower in PPI users (12%) than in P-CAB users (68%). In general, P-CAB more strongly suppresses gastric acid and further increases the serum gastrin level compared with PPIs. The WS-positive group had a significantly greater percentage of patients with a high serum gastrin level and a significantly greater percentage of P-CAB users than the WS-negative group. Although the mechanism by which hypergastrinemia in P-CAB users induces WS was not revealed in the present study, there was a definite relationship between P-CAB usage and the presence of WS. Since A-SA generally increases the gastrin level, whether gastrin or A-SA is directly associated with WS prevalence remains unclear. Moreover, two patients without both A-SA uptake and high gastrinemia had WS prevalence in the control group (Table 1). Further investigations are needed to reveal these reasons. Furthermore, there was no difference in the duration of A-SA administration, whereas a minimal time of WS prevalence was 1 month after A-SA uptake. Our previous report also described that WS disappeared after discontinuation of P-CAB for 2 months [5]. Therefore, WS is probably a reversible A-SA-associated finding that appears and disappears in a relatively short term.

The histological findings of a previous study showed that GCM is cystic gland dilation in 79.2% (19/24), PCP in 75% (18/24), and PCV in 29.2% (7/24) [17]. Furthermore, black spot reportedly accounts for 67.6% (23/34) of dilation, and 76.5% (26/34) of PCP [14], while there are no data on PCV. The present study revealed that WS was present in 43% (6/14) of patients with cystic gland dilation, 71% (10/14) of patients with PCP, and 86% (12/14) of patients with PCV. The differences between the endoscopic findings in the present study versus the two abovementioned previous studies may be because the antacid in the previous studies was a PPI, while the present study included P-CAB, which more strongly suppresses gastric acid than PPIs.

PCP describes the marked convex protrusion of parietal cell cytoplasm into the lumen of oxyntic glands. Oxyntic glands affected by PCP tend to dilate (oxyntic dilatation), and the cytoplasm of the involved parietal cells often appears vacuolated (PCV) [18]. In a previous study, the prevalence of PCP increased during omeprazole therapy from 18% at baseline to 79% and 86% at 3 and 12 months, respectively [19]. Furthermore, the administration of omeprazole reportedly causes vacuolation in approximately 27% of all parietal cells [20]. Our histological analysis revealed that P-CAB usage induced both PCP and PCV. Thus far, there have been many reports of PPIs causing PCP, but there are no pathological reports of P-CAB causing PCP. Our data showed that WS, as a P-CAB-related lesion, may suggest the presence of severe gastric mucosal abnormalities with PCP and PCV.

Previous studies have shown that the blood gastrin level is significantly higher in patients taking P-CAB than in those taking a PPI [21,22]. However, the presence of GCM is not related to gastrin levels [15]. Some studies report that FGP is related to gastrin levels, while others report that they are not [12,23]. The present study revealed a strong association between the presence of WS and serum gastrin levels. In addition, among the patients using P-CAB, the WS expression rate was significantly higher in the patients with a high serum gastrin level than in those without a high serum gastrin level; thus, the present result was slightly different from the endoscopic findings in other PPI-related gastric disorders. As shown in Table 1, there were significant differences regarding age and presence of WS in between control and A-SA group. Thus, we additionally performed a multivariate analysis regarding WS prevalence. As the result, age was not detected as a risk factor to develop WS.

Some possible mechanisms can be hypothesized for the pathogenesis of WS. First, P-CAB leads to hypergastrinemia through its trophic action, and excessive bifurcation from the isthmus of the gland to the neck of the gland also causes structural disturbance and obstructs outflow from the gland, finally resulting in cystic gland dilatation. Second, WGA occurs when the turnover of epithelial cells is increased in cancer, or epithelial turnover is excessively increased in hypergastrinemia, and this causes necrotic atypical epithelium. We speculate that necrotic tissue in the cystic gland is observed as WS, similarly to WGA.

The findings in autoimmune gastritis suggest that WS is related to parietal cell degeneration and high serum gastrin levels [3], which are common conditions. In the present study, there were many older adults in the A-SA group; therefore, P-CAB usage and high serum gastrin level, as well as older age, might be a factor related to the appearance of WS and black spots.

With the emergence of P-CAB and the increasing frequency of its use, the prevalence of WS is expected to increase in the future. The presence of WS may suggest that the patient has hypergastrinemia, and therefore, clinicians should try changing or reducing the dose of A-SA.

### Study Limitations

The present study was a single-center observational study performed in a facility with many older adult patients. Patients using other antacids (including H2-blockers) except for PPIs and P-CAB were not excluded in the control group because our previous experience has shown that no WS appeared in patients taking these antacids. At this point, there was a small bias. Detailed information including serum gastrin and *H.pylori* antibody of patients from the control group was insufficient because the blood testing was dependent on each patient’s discretion. Therefore, other causative factors of WS could not be analyzed in the control group. Although age did not result in a risk factor of WS prevalence in this study, study cohorts adjusted to age should be designed to prove the relationship between age and WS prevalence. The present findings require confirmation in a future multicenter large-scale trial.

## 5. Conclusions

WS may be a novel endoscopic finding of A-SA-related gastric lesions. The presence of WS may be associated with P-CAB use and hypergastrinemia.

## Figures and Tables

**Figure 1 jcm-10-02625-f001:**
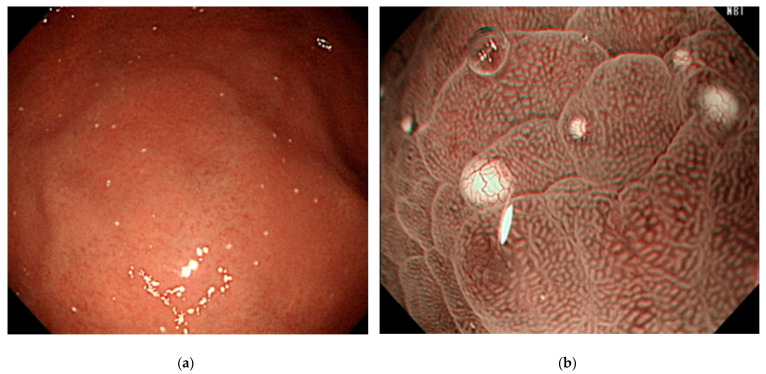
Endoscopic images: (**a**) multiple, round, slightly elevated, white substances are observed on white-light imaging; (**b**) magnifying narrow-band imaging reveals the presence of microvessels overlying the white lesion with a globular shape.

**Figure 2 jcm-10-02625-f002:**
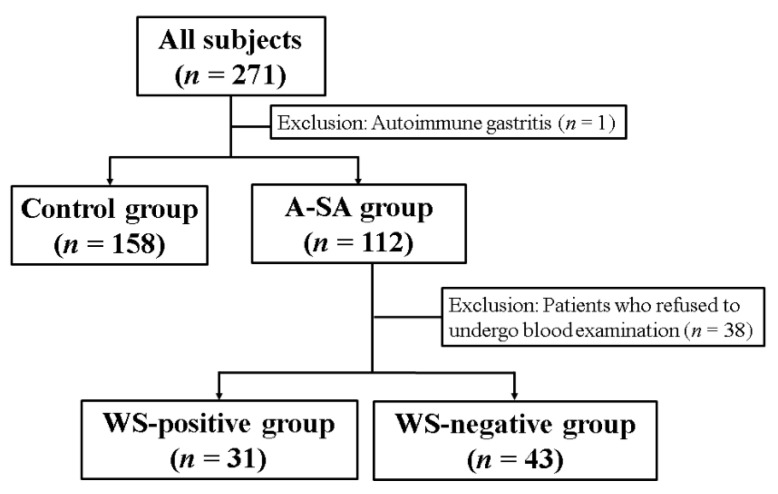
Flowchart of the present study: A-SA, acid-suppressing agents; WS, white spot.

**Figure 3 jcm-10-02625-f003:**
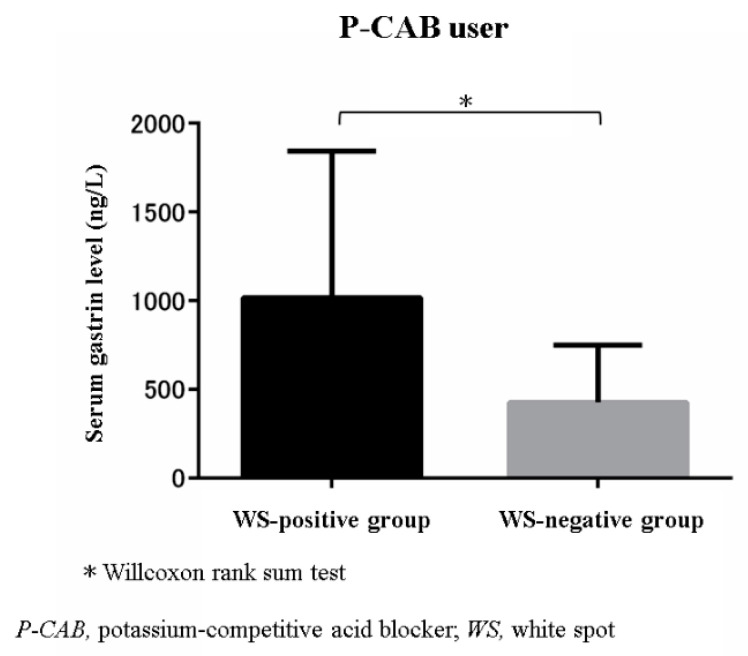
Comparison of the fasting serum gastrin level between the WS-positive and WS-negative groups within the group of patients using P-CAB: * *p* < 0.05; Wilcoxon rank-sum test; WS, white spot; P-CAB, potassium-competitive acid blocker.

**Figure 4 jcm-10-02625-f004:**
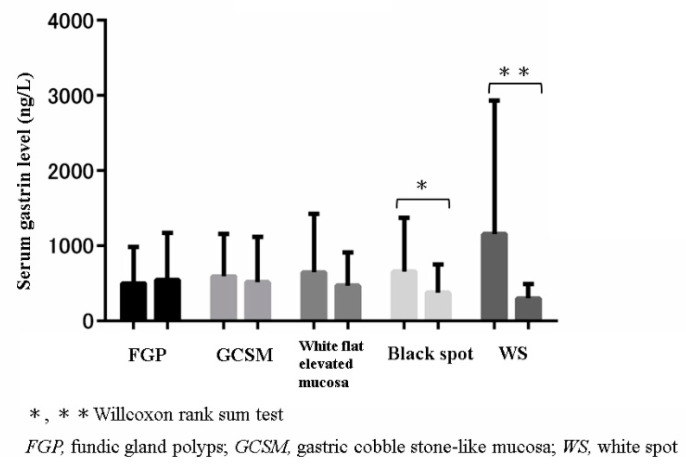
Comparison of the fasting serum gastrin level between patients with and without each PPI-related gastric lesion: * *p* < 0.05; ** *p* < 0.01; Wilcoxon rank-sum test; PPI, proton pump inhibitor; FGP, fundic gland polyps; GCSM, gastric cobblestone-like mucosa; WS, white spot.

**Figure 5 jcm-10-02625-f005:**
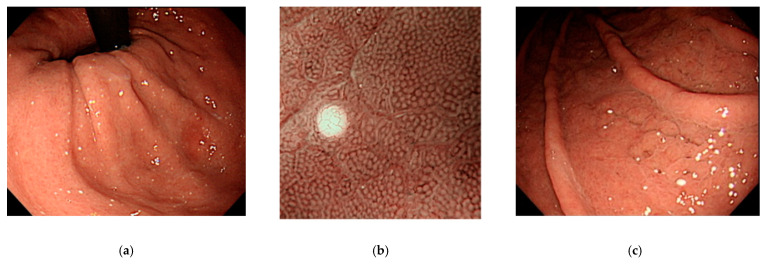
Endoscopic images of white spot: (**a**) multiple white spots at the fornix of the stomach; (**b**) nonatrophic mucosa with a round pit surrounded by white spot on magnifying narrow-band imaging; (**c**) gastric-cracked mucosa (a proton-pump-inhibitor-related lesion) in a patient with white spot.

**Figure 6 jcm-10-02625-f006:**
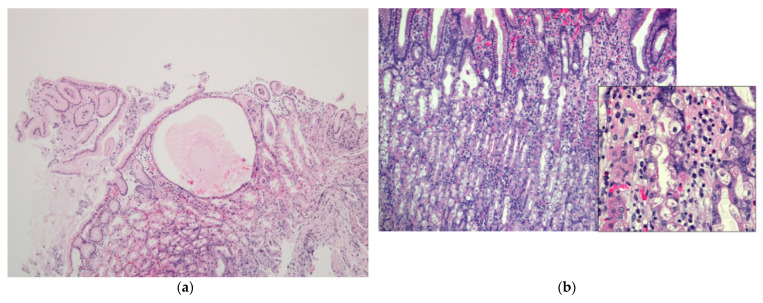
Histopathological findings: (**a**) a dilated gland that contains eosinophilic material with necrotic epithelial fragments; (**b**) parietal cell protrusion and parietal cell vacuolation in the mucosa surrounding a white spot in the fundic gland area.

**Table 1 jcm-10-02625-t001:** Characteristics of the 270 patients in the control and A-SA groups.

	Control Group (*n* = 158)	A-SA Group (*n* = 112)	*p* Value
Age (years), mean ± SD	61.8	78.7	<0.001 ^#^
Sex, *n* (%)			0.209 *
Male	57 (36.0)	49 (43.7)	
Female	101 (63.9)	63 (56.2)	
Presence of WS, *n* (%)	2 (1.3)	31 (27.6)	<0.001 ^#^

A-SA, acid-suppressing agents; WS, white spot; ^#^ Wilcoxon rank-sum test; * Fisher’s exact test.

**Table 2 jcm-10-02625-t002:** Prevalence of white spot in PPI users versus P-CAB users within the A-SA group.

	PPI Users (*n* = 81)	P-CAB Users (*n* = 31)	*p* Value
Presence of WS, *n* (%)	10 (12)	21 (68)	<0.001 ^&^

PPI, proton pump inhibitor; P-CAB, potassium-competitive acid blocker; WS, white spot; ^&^ Fisher’s exact test.

**Table 3 jcm-10-02625-t003:** Characteristics of the 74 patients in the WS-positive and WS-negative groups.

	WS-Positive Group (*n* = 31)	WS-Negative Group (*n* = 43)	*p* Value
Age (years), median (range)	82 (57–94)	82 (59–99)	0.6487 ^(1)^
Sex			0.8790 ^(2)^
Male	11	16	
Female	20	27	
*H. pylori* antibody			0.0176 ^(3)^
<10 U/mL	31	35	
≥10 U/mL	0	8	
*H. pylori* eradication history	1	1	1.000 ^(3)^
Renal function, eGFR			0.4618 ^(2)^
G1–G3a	24	30	
G3b–G5	7	13	
Serum gastrin (pg/mL), median (range)	624 (95–10,082)	261 (73–954)	<0.001 ^(4)^
High gastrin level (≥500 pg/mL)	18	6	<0.001 ^(2)^
Duration of A-SA use (years)			0.282 ^(2)^
≤1	7	6	
1–3	11	11	
≥3	13	26	
A-SA type			<0.001 ^(3)^
PPI	10	38	
P-CAB	21	5	

WS, white spot; eGFR, estimated glomerular filtration rate; PPI, proton pump inhibitor; P-CAB, potassium-competitive acid blocker; ^(1)^ Student’s t-test; ^(2)^ Pearson’s chi-square test; ^(3)^ Fisher’s exact test; ^(4)^ Wilcoxon rank-sum test.

**Table 4 jcm-10-02625-t004:** Multivariate analysis of WS prevalence.

	Odds Ratio	95%CI	*p* Value
Age (1-year increments)	1.096	0.220–5.466	0.9103
Gender (Male)	1.034	0.209–5.110	0.9667
*H.pylori* eradication history	5.122	0.232–112.7	0.3003
Renal function, eGFR (G3b–G5)	0.946	0.180–4.975	0.9483
High gastrin (≧500 pg/mL)	6.099	1.291–28.81	0.0224
Duration of A-SA use (years, 3≦)	0.732	0.164–3.268	0.683
P-CAB user	25.73	4.318–153.34	0.0004

WS, white spot; eGFR, estimated glomerular filtration rate; P-CAB, potassium-competitive acid blocker.

**Table 5 jcm-10-02625-t005:** Endoscopic findings in the 33 patients with white spot.

Endoscopic Finding	Number (%)
WS in the upper gastric body	26 (79)
WS in the middle gastric body	16 (48)
WS in the lower gastric body	8 (24)
WS in the antrum	1 (3)
Presence of multiple WSs	29 (88)
Atrophy surrounded by WS	2 (6)
PPI-related gastric lesions accompanying WS	
Black spot	23 (70)
Fundic gland polyp	5 (15)
White flat elevated mucosa	15 (46)
Cobblestone-like mucosa	9 (27)
Hyperplastic polyp	2 (6)

WS, white spot.

**Table 6 jcm-10-02625-t006:** Histopathological findings of the 14 patients from whom a biopsy of a white spot area was obtained.

Histopathological Finding	Number (%)
Parietal cell protrusion	10 (71)
Parietal cell vacuolation	12 (86)
Cystic gland dilation	6 (43)

## Data Availability

The data presented in this study are available on reasonable request from the corresponding author

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
