# Peer review of "White Spot, a Novel Endoscopic Finding, May Be Associated with Acid-Suppressing Agents and Hypergastrinemia"

_jcm, 2021, doi:10.3390/jcm10122625_

Round 1
Reviewer 1 Report
The authors describe the association of endoscopic white spots to the intake of antacids (AA). The paper is clearly written and interesting. There are some minor points that should be addressed:
- Why the control cohort was not exclusively recruited from patients without any AA? Even with the explanation of the authors this is a small bias.
- Regarding point 1, it would be interesting if the 2 patients with WS in the control cohort did take any AA?
- Age may be a risk to develop WS. There was a significant difference. Can't the cohorts be adjusted to age? If not, this point must be discussed clearly and added to the limits of the study!
- Using AA increases gastrin levels in general. Therefore it is not really astonishing that both correlate with WS. Have patients be indentified with high gastrin, no AA and ES in your experience? Are WS really gastrin associated or are they AA associated? Does it play a role how long the AA are taken? Is there a minimal time of AA intake before WS occur? These points must be discussed.
Author Response
To,
Editor-in-Chief & Co-Editor, Journal of Clinical Medicine
Dear Professors
Thank you very much for your letter with regard to our manuscript entitled " White Spot, a Novel Endoscopic Finding, may be Associated with Acid-Suppressing Agents and Hypergastrinemia". (Manuscript ID: jcm-1250076)
We are grateful for the provided feedback and have revised our manuscript accordingly. Our response to the referee comments is provided on the following pages and we have attached our revised manuscript; two versions of word documents.
{track changed manuscript}, and {clean manuscript}.
We hope that the revised manuscript is now suitable for publication in Journal of Clinical Medicine
.
Your kind and timely consideration of our article would be greatly appreciated.
Sincerely,
Noriko Nishiyama, MD, PhD
Response to Reviewer1 comments:
- Why the control cohort was not exclusively recruited from patients without any AA? Even with the explanation of the authors this is a small bias.
Response
Thank you for your valuable comments.
We already mentioned in the 2.1. Study Design, Setting and Participants as follows;
Patients using other antacids (including H2-blockers) were included in the control group because these agents are not strong A-SAs and our previous experience has shown that no WS appears in patients taking these antacids.
Therefore, the control cohort was not exclusively recruited from patients without any antacids. As you mention, there is a small bias. Thus, we added the following sentence in the limitation of discussion ‘Patients using other antacids (including H2-blockers) except for PPIs and P-CAB were not excluded in the control group because our previous experience has shown that no WS appeared in patients taking these antacids. In this point, there was a small bias.’
- Regarding point 1, it would be interesting if the 2 patients with WS in the control cohort did take any AA?
Response
As you comment, it would be interesting.
We added the following sentence at discussion session.; ‘Moreover, two patients without both A-SA uptake and high gastrinemia had WS prevalence in the control group (Table 1). Further investigations are needed to reveal these reasons.’
- Age may be a risk to develop WS. There was a significant difference. Can't the cohorts be adjusted to age? If not, this point must be discussed clearly and added to the limits of the study!
Response
Thank you for your valuable comments.
We added the following sentence in the discussion; ‘As shown in Table 1, there were significant differences regarding age and presence of WS in between control and A-SA group. Thus, we additionally performed a multivariate analysis regarding WS prevalence. As the result, age was not detected as a risk factor to develop WS.’
Moreover, we added the following sentence in Study Limitations;‘Although age did not result in a risk factor of WS prevalence in this study, study cohorts adjusted to age should be designed to prove the relationship between age and WS prevalence.’
- Using AA increases gastrin levels in general. Therefore it is not really astonishing that both correlate with WS. Have patients be indentified with high gastrin, no AA and ES in your experience? Are WS really gastrin associated or are they AA associated? Does it play a role how long the AA are taken? Is there a minimal time of AA intake before WS occur? These points must be discussed.
Response
Thank you for your precise comments.
As you address, we already mentioned in the discussion as follows; The WS-positive group had a significantly greater percentage of patients with a high serum gastrin level and a significantly greater percentage of P-CAB users than the WS-negative group. Although the mechanism by which hypergastrinemia in P-CAB users induces WS was not revealed in the present study, there was a definite relationship between P-CAB usage and the presence of WS.
I added the following sentences in the discussion; ‘Since A-SA generally increases the gastrin level, whether gastrin or A-SA is directly associated with WS prevalence remains unclear. Moreover, two patients without both A-SA and high gastrinemia had WS prevalence in the control group. Further investigations are needed to reveal these reason. Furthermore, there was no difference in the duration of A-SA administration. Whereas, a minimal time of WS prevalence was 1 month after A-SA uptake. Our previous report also described that WS disappeared after discontinuation of P-CAB for 2 months [5]. Therefore, WS is probably a reversible A-SA associated finding that appears and disappears at relatively short-term.’
Reviewer 2 Report
A very interesting study. In my opinion, it requires minor revisions.
1- If one of the factors differentiating the group is age, single and multivariate regression analysis should be presented in order to exclude the influence of age as a causative factor of endoscopic changes - please present the relevant analysis.
2- Please provide details of patients from control group with WS. Did they present hyperergastrinemia? Could any other factors have a potential influence on the incidence of WS in this group? Have these patients used PPIs or other inhibitors previously?
Author Response
To,
Editor-in-Chief & Co-Editor, Journal of Clinical Medicine
Dear Professors
Thank you very much for your letter with regard to our manuscript entitled " White Spot, a Novel Endoscopic Finding, may be Associated with Acid-Suppressing Agents and Hypergastrinemia". (Manuscript ID: jcm-1250076)
We are grateful for the provided feedback and have revised our manuscript accordingly. Our response to the referee comments is provided on the following pages and we have attached our revised manuscript; two versions of word documents.
{track changed manuscript}, and {clean manuscript}.
We hope that the revised manuscript is now suitable for publication in Journal of Clinical Medicine
.
Your kind and timely consideration of our article would be greatly appreciated.
Sincerely,
Noriko Nishiyama, MD, PhD
Response to Reviewer2 comments:
- If one of the factors differentiating the group is age, single and multivariate regression analysis should be presented in order to exclude the influence of age as a causative factor of endoscopic changes - please present the relevant analysis.
Response
Thank you for your valuable comments.
In this time, we performed a multivariate analysis of WS prevalence, and we added the following sentence in Statistical Analysis;
A multivariate analysis was performed using multiple logistic regression to detect a causative factor of WS prevalence.
We also added the results and Table 4 as follows;
Multiple Logistic Regression Analysis of WS prevalence was shown in Table 4.
The multivariate analysis revealed that high gastrin and P-CAB user were independent causative factor of WS prevalence with odds ratio of 6.099 (p=0.0224) and 25.73 (p=0.0004), relatively.’
Age was not detected as a risk factor to develop WS (P; 0.9103).
2. Please provide details of patients from control group with WS. Did they present hyperergastrinemia? Could any other factors have a potential influence on the incidence of WS in this group? Have these patients used PPIs or other inhibitors previously?
Response
I am sorry it’s complicated.
We could not take detailed information including serum gastrin and H.pylori antibody of patients from control group because the blood testing was dependent on each patient’s discretion, some patients did not provide informed consent for blood testing. Therefore, other causative factors of WS could not be analyzed in the control group. None of patients in the control group had a previous history with PPIs or P-CAB.
We added the following sentence in the limitation; Detailed information including serum gastrin and H.pylori antibody of patients from control group was insufficient because the blood testing was dependent on each patient’s discretion. Therefore, other causative factors of WS could not be analyzed in the control group.
Others
We are calculated p-values in Table 3 using the latest version, JMP 15.1.0 (SAS Institute Inc., Cary, NC, USA).